# Dietary diversity and associated factors among women of reproductive age in Jeldu District, West Shoa Zone, Oromia Ethiopia

**Gudisa Merga[1], Samson Mideksa[2], Nagasa Dida[3]\*, Gina Kennedy[4]**

**1** West Shewa Zonal Health Office, Ambo, Oromia, Ethiopia, **2** Ethiopian Public Health Research Institute, Addis Ababa, Ethiopia, **3** Public Health Department, College of Medicine and Health Sciences, Ambo University, Ambo, Oromia, Ethiopia, **4** Global Alliance for Improved Nutrition, Wageningen University, Wageningen, Netherlands

\* nadibefe@gmail.com

**Data Availability Statement:** All relevant data are within the article and its Supporting Information files.

## Abstract

### Background

Women of reproductive age are at a higher risk of insufficient micronutrient intake due to their low dietary diversity which has an impact on child growth and development, anemia and low birth weight. However, there are no information from the study area. Hence, the study aimed to assess dietary diversity and associated factors among reproductive-age women in Jeldu District, West Shoa Zone, Oromia Region, Ethiopia, 2018.

### Methods

A community-based cross-sectional study was undertaken among 634 women of reproductive age. The study participants were recruited using a systematic sampling method. The data were collected using a structured questionnaire based on 24-hour dietary recalls. The data were checked, coded, and entered into EpiData version 3.1 before being exported to SPSS Version 21 for analysis. Descriptive statistics like frequency standard deviation, mean and proportions were computed., both binary and multivariable logistic regressions were run at 95 percent confidence intervals. A P-value of <0.05 was used to declare a statistically significant association between dietary diversity and explanatory variables.

### Results

The proportions of women who consumed greater than or equal to five food groups were 81.9%. Agro-ecological zone of highland (AOR = 7.71: 95% CI: 3.72, 15.99), women who have a radio (AOR; 1.87: 95% CI; 1.17, 2.99) and women's decision-making power to purchase food for household (AOR; 3.93:95% CI; 2.3, 6.71) and having own mobile phone (AOR: 1.92 (1.74, 3.16)) were statistically associated with food dietary diversity.

### Conclusion

The proportion of women who met the minimal standard for dietary variety requirements was high. The presence of radios, mobile phones, women's purchasing decision power, as

**Funding:** Consultative Group on International Agricultural Research (CGIAR) from a Research Program on Agriculture for Nutrition and Health (A4NH) under the flagship of Food Systems for Healthier Diets (FSHD).

**Competing interests:** The authors declare that they have no competing interests.

**Abbreviations:** AOR, Adjusted Odds Ratio; CGIAR, Consultative Group on International Agricultural Research; CI, Confidence Interval; COR, Crud Odds Ratio; DDS, Dietary Diversity Score; FAO, Food and Agriculture Organization; MDD-W, minimum dietary diversity for women; SPSS, Statistical Package for Social Sciences; WRA, women of reproductive age.

well as possessing large cattle, and the agroecological zone of the participants were all important predictors of dietary variety among reproductive-age women. The local media, agriculture office, health office, and women, youth, and children office all need to pay more attention to the determinants of dietary variety in women.

## Background

Dietary diversity refers to the variety of foods consumed across and within food categories within a certain period, as well as a rise in the range of foods available across and within food groups that can ensure optimal intake of key nutrients for good health [1].

Dietary diversity is typically measured by adding up how many different foods were taken over some time or, more commonly, by counting how many different food groups were consumed [2, 3]. The proportion of women aged 15 to 49 who consume at least 5 out of 10 food groups is known as minimum dietary diversity for women (MDD-W). It is an indicator of women's diet quality with a focus on micronutrient sufficiency and a measure of household access to a diet high in micronutrients [4, 5].

All people must consume a variety of foods to meet their nutritional needs; a diverse diet is important and helps to ensure optimal intake of micronutrients; No one food has all the nutrients; The possibility of fulfilling nutrient needs is higher when more food types are consumed daily [2, 5, 6]. Low dietary intakes and unfair intra-household food distribution are the leading causes of undernutrition, and food taboos and misconceptions contribute significantly to the high levels of undernutrition among vulnerable groups [7–10].

Most monotonous staple meals are deficient in important micronutrients, resulting in micronutrient deficits in susceptible groups (especially women of reproductive age) [11]. Many people in underdeveloped nations lack the resources to cultivate or purchase micronutrient-dense foods like meat, fish, poultry, eggs, milk, and dairy products, and their diets are based on nutrient-deficient staples like rice and maize [12].

In underdeveloped nations such as Africa, Asia, Latin America, and the Caribbean, pregnant and breastfeeding women who ate mostly plant-based diets had insufficient micronutrient intakes and minimal dietary diversification [13]. Micronutrient deficiency, also known as hidden hunger, affects more than two billion people worldwide. Key micronutrient deficiencies, such as iron, iodine, vitamin A, folate, vitamin D, and zinc, can have serious health consequences in young children, adolescent girls, pregnant, and lactating women [14].

Annually, maternal malnutrition causes 800,000 neonatal deaths;, stunting, wasting, and maternal micronutrient deficiencies cause approximately 3.1 million child deaths [15, 16]; all forms of malnutrition are caused by poor diets, insufficient knowledge and resources, and unhealthy environments; if current trends continue, one person in two will be malnourished by 2030 [17]; poor maternal nutrition is directly associated with mothers' lack of resistance to infection and wasting; poor maternal nutrition is directly associated with mothers' lack of resistance [18]. Due to their poor dietary diversity, Ethiopian women of reproductive age are at a higher risk of micronutrient deficiency [19]. But, there is no information on dietary divericity practice among reprocuttive age women of Jeldu District. As a result, this study assessed dietary diversity practice and its associated variables among women of reproductive age in Jeldu district, West Shoa zone, Oromia Ethiopia, in 2018.

## Method and materials

### Study design and setting

A community-based cross-sectional study was conducted from March to April 2018 in Jeldu district, West Shoa zone, Oromia Ethiopia.

A single population proportion formula was used to estimate the sample size. The sample size was calculated using the following assumptions: 50% as prevalence of adequate dietary diversity among women of reproductive age, 95% confidence level, 1.5 design effects, 10% nonresponse rate, and 5% degree of precision. Finally, a sample size of 634 was achieved.

The district was stratified into agroecological zones and then ten sub-districts (the smallest administration units) were selected by simple random sampling from the total thirty subdistricts by lottery method.

### Sample size and sampling technique

Households were selected using systematic random sampling; a community health information system (CHIS) was used as a sampling frame. Then six hundred thirty-four (634) samples were proportionally allocated for each selected subdistrict. Finally, one woman was selected using a lottery method if more than one woman presents in a household. Tool and Data Collection Methods: Regarding data collection semi-structured questionnaire was used to collect data tool was adapted after reviewing different literatures, food and Agriculture Organization (FAO); FHI 360 (1, 4). Minimum Dietary Diversity for Women and health survey reports. First, the tool was prepared in English and translated to Afan Oromo, and re-translated to English by language experts to check the consistency of the questionnaire.

Thirteen (13) data collectors who speak the Afan Oromo language were trained for two days before the actual data collection on interviewing approach, data collection technique, objectives of the study, participant's rights, techniques of data collection, and completeness of the data. The questionnaire was pretested among thirty-two households out of the study area (Edensa Gelan).

The outcome, dietary diversity, was assessed using Food and Agricultural Organization to measure women's dietary diversity. Food consumed by women of reproductive age was assessed through the 24-hour recall method (food consumed in one day from the time of interview back) and then food items were categorized into ten food groups. Dietary Diversity Score (DDS) was created as a summary measure of dietary intake, accordingly, participants who had DDS of five and above were considered as having adequate dietary diversity, whereas inadequate DDS was determined when they had less than five DDS [4, 21].

### Operational definition

In this study, large-Livestock mean animal like—cattle, horses, donkey, goats, and sheep; whereas small livestock is animals like hen and cock; Adequate dietary diversity was assessed using the ten FANTA food groups. Inadequate dietary diversity: when women of reproductive age consumed less than five of ten food groups within 24 hours; Adequate dietary diversity: when reproductive women age consumed at least five of ten food groups within 24 hours.

### Data management and analysis

Epi-Data 3.1 was used for data entry and exported to SPSS Version 21 window for analysis. Descriptive statistics like frequency and percentage for the categorical variables, and mean and standard deviation for continuous variables were applied. Binary logistic regression analysis was done to identify candidate variables for multivariable logistic regression model and

variables having a p ≤ 0.20 were candidates for multivariable logistic regression and entered into multivariable logistic regression model. Variables with less than (p ≤ 0.05) were declared statistically significant at 95% of the confidence interval.

### Ethics approval and consent to participate

This study was approved by the ethical review board of the College of Medicine and Health Sciences, Ambo University. The ethical review board also approved its ethical issues as there was no procedure that affects the study subject and the data is used only for research purpose. Hierarchically all administrative bodies were communicated and permission was secured. Informed consent to participate in the study was obtained from all participants after explaining objectives and procedures of the study and their right to participate or to withdraw at any time in between of the interview. For this purpose, a one-page consent letter was attached to the cover-page of each questionnaire stating about the general purpose of the study and issues of confidentiality which was discussed by data collectors before proceeding to the interview. Lastly, we confirm that this study was conducted in accordance with the Declaration of Helsinki.

## Results

### Socio-demographic economic and characteristics

A total of 634 women of reproductive age participated in this study, making a response rate of 100% with a mean age of (30.55) years and a standard deviation of 7.628. About 44.2% of participants were from the Highlands, 36.9% from the Midlands, and 18.9% from the Lowlands. More than (53.6%) of the respondents were protestant in religion, about 83% were married, and 55.2% of the participants had no formal education. 57.1% of respondents were farmers, and 50.6% of participants lived with less than five family members (Table 1).

Regarding average monthly income, more than half (62.1%) of women earned 501–1500 Ethiopian birr (Average exchange rate in 2018:1$ = 27.6677 ETB). More than three-fourth (77%) women were doing home gardening and about 83% households had farmland. Regarding the communication material, more than half (56.9%) and two-thirds (73%) of participants had radio and mobile phone, respectively (Table 2).

### Food dietary diversity and associated factors

Overall, 81.9% (95% CI: 78.9–84.9) women had adequate dietary diversity. Moreover, the mean dietary diversity score of participants was 5.98 (SD± 1.86). The majority (99.8%) of women consumed starchy staples (grains, roots, and tubers). However, only 24.1%ate dark green leafy vegetables (Fig 1).

Using bivariable logistic regression analysis, the variables identified at a p-value of < 0.2 for the final model were age, agro-ecological zone, marital status, having a bank account at any bank, having home gardening, having farmland, having small animals, having large animal, average monthly income, radio usage, mobile phone usage, food exchange, main source of food from market, main source of food own production, women's decision making to purchase food, and animal used as means of transportation. In multivariable logistic regression, variables like women's decision-making power to purchase food for the household, Agro-ecological zone, having large livestock, having a radio, and having a mobile phone were significantly associated with dietary diversity. Women in the highland were 7.71 (AOR = 7.71; 95% CI: 3.72–15.99) more likely to consume adequate dietary diversity than women of reproductive age in the lowlands. Women who had radio were 1.87 times more likely to have adequate

**Table 1. Socio-demographic characteristics of reproductive women in Jeldu district West Shoa zone, Oromia, Ethiopia, March 25 –April 15, 2018.**

| Variables | Category | Frequency | Percentage |
|---|---|---|---|
| Age | 15–24 | 135 | 21.3 |
| | 25–34 | 282 | 44.2 |
| | 35–44 | 188 | 29.7 |
| | ≥45 | 29 | 4.6 |
| Agro-ecological Zone | Highland | 280 | 44.2 |
| | Midland | 234 | 36.9 |
| | Lowland | 120 | 18.9 |
| Marital status | Single | 81 | 12.8 |
| | Married | 526 | 83 |
| | Divorce | 17 | 2.7 |
| | Widowed | 10 | 1.6 |
| Religion | Orthodox | 244 | 38.5 |
| | Waqefta | 50 | 7.9 |
| | Protestant | 340 | 53.6 |
| Educational status | No formal education | 350 | 55.2 |
| | Primary | 241 | 40.1 |
| | Secondary and above | 43 | 6.8 |
| Family size | <5 | 321 | 50.6 |
| | ≥5 | 313 | 49.4 |
| Occupational status | House wife | 141 | 22.2 |
| | Farmers | 362 | 57.1 |
| | Students | 89 | 14 |
| | *Others | 42 | 6.6 |

*Merchant: government employments and daily labors.

diversified food (AOR = 1.87; 95%; CI: 1.17–2.99) compared to their counterparts. Women with purchasing power for household food were 3.93 times more likely to consume diversified food (AOR = 3.93; 95% CI:2.3–6.71) as compared to women with no purchasing power., Women who had large livestock were 2.67 times more likely to have adequate diversified food compared to those who women who had no large livestock (AOR; 2.67: 95% CI: 1.4–5.08) and women who had mobile phones were 1.92 times more likely to consume adequate dietary diversity (AOR = 1.92; 95% CI: 1.74–3.16,) compared to their counterparts (Table 3).

## Discussion

This study was conducted among reproductive age women of Jeldu District to determine ditery divercity practice and its associated factors. Accordingly, this study revealed that more than three-fourths (81.9%) of women consumed the minimum recommended food groups or has got adequate dietary diversity. Compared to other study findings, it is higher than the findings of: South Africa 25% [20], India, 46.2% to 76.6% [21, 22], Algeria, 32% [23], Bangladesh 65% [24] and studies in other part of Ethiopia, 28% to 50.7% [25–27] however it is lower than the study conducted in Zambia 87.5% [28]. The high proportion of adequate dietary diversity in this study might be due to the nature of the study, i.e, the study period. Moreover, the study area is known district of tuber and grain producers. This study also confirmed that almost all (99.8%) women consumed grains, roots, and tubers. This result was supported by the study conducted in Ghana, Zambia; in Kenya and other area of Ethiopia [28–30].

**Table 2. Socio-economic characteristics in Jeldu district West Shoa zone, Oromia, Ethiopia, March 25 –April 15, 2018.**

| Variables | Category | Frequency | Percent |
|---|---|---|---|
| Head of household(n = 634) | Male | 538 | 84.9 |
| | Female | 96 | 15.1 |
| HH member have saving account at any bank | Yes | 171 | 27 |
| | No | 463 | 73 |
| Average monthly Income (Ethiopian birr) | ≤500 | 123 | 19.4 |
| | 501–1500 | 394 | 62.1 |
| | ≥1501 | 117 | 18.5 |
| Production grow on farmland(n = 526) | Wheat | 349 | 66.3 |
| | Teff | 327 | 62.2 |
| | Barely | 251 | 47.7 |
| | Sorghum | 173 | 32.9 |
| | Maize | 255 | 48.5 |
| | Onion | 60 | 11.4 |
| | Chickpea | 73 | 13.9 |
| | Potatoes | 331 | 62.9 |
| | Bean | 44 | 8.4 |
| | Peas | 20 | 3.8 |
| Having mass media or having electronics device in the HH | Having radio | 361 | 56.9 |
| | Having mobile | 463 | 73 |
| | Having TV | 59 | 9.3 |

The result of multivariable logistic regression analysis showed that women who were living in the highlands were 7.71 times more likely to consume adequate dietary diversity when compared with those who were living in the lowlands. This result agreed with the study conducted in the Limu district of Hadiya Zone, Southern Ethiopia [31]. The similarity could be due to the high production of different grains and tubers in the highlands and the availability of varieties of crops in the highland, moreover presence of many market days and shops could avail availability of healthier diets like vegetables and fruits in both studies, and almost all in highland areas vegetables like potatoes exist and are consumed throughout the year. But inconsistent with the study conducted in Raya Alamata, Southern Tigray, Ethiopia [18, 32].

The difference is due to availability of the production, and socio-cultural determinants which might need further concerns. The community accessibility to market supplies that population resides near to market could have better access to a variety of foods and the agro-ecological zone could play great role for the place where these market exit.,; these could be risk factors for a reduction in agricultural productivity when a variety of products are not available an individual cannot consume varieties of food groups. Women who had radio were 1.87 times more likely to consume adequate dietary diversity when compared with their counterparts. This result is consistent with a study conducted in Ethiopia [33, 34].

The similarity is due to access to information through broadcasts of nutrition and health messages as a means of advocacy, and it is related to their economic status; women with higher monthly income can buy mobile and radio services and receive nutrition massages. Women who owned large livestock were 2.67 times more likely to consume adequate dietary diversity. Studies in rural Bangladesh and other parts of Ethiopia are congruent with this findings

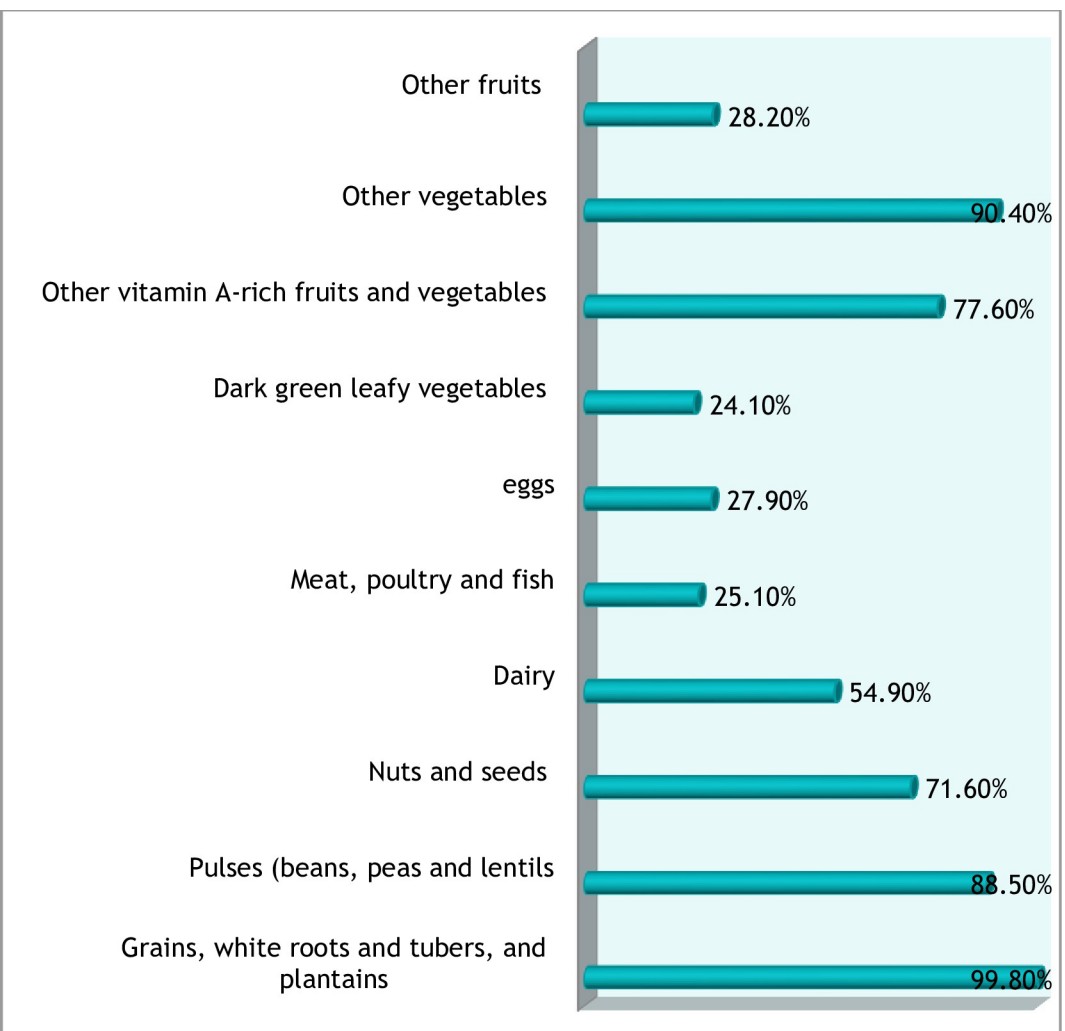

**Fig 1. Food groups consumed among women reproductive age in Jeldu West Shoa Zone Ethiopia, 2018.**

[35, 36]. This consistent association suggests that households that own large livestock are more likely to move from inadequate dietary diversity to adequate dietary diversity and are easy to keep, trade, and contain foods that may provide micro- and macronutrients when their products are consumed, and also used as income. Lastly, women who have the decision-making power to purchase food for their household were 3.93 times more likely to consume adequate dietary diversity, which is consistent with the study in Nepal and the Netherlands, Bangladesh, Ghana, and Ethiopia [29, 33, 37–41].

The study tried to show the dietary diversity of one of the most vulnerable groups of the population on behalf of the Jeldu district, West Shoa, Oromia Ethiopia and, but some of the limitations of this study should be taken into consideration. This study did not consider the quantity of food consumed by women of reproductive age and a single twenty-four-hour recall did not indicate the normal dietary routine of women. There might be social desirability bias and recall bias during data collection which do not show cause and effects. The study period would not represent reality exist in the community across the year. Moreover, it would have been good if the study was supported by qualitative study to address different social-cultural food taboos in the community.

**Table 3.  Binary and multivariable logistic regression model showing socio demographic, socio economic and personal food environment factors associated with dietary diversity among reproductive age women in Jeldu district West Shoa zone, Oromia, Ethiopia, March 25 –April 15, 2018.**

| Variables | Category | WRA dietary diversity | | COR(95%CI) | AOR(95%CI) | p-value |
| --- | --- | --- | --- | --- | --- | --- |
| | | Adequate | Inadequate | | | |
| Age | 15–24 | 111(82.2%) | 24(17.8%) | 2.83(1.18–6.75) | 2.07(0.75–5.73) | 0.158 |
| | 25–34 | 131(81.9%) | 51(18.1%) | 2.77(1.23–6.22) | 2.18(0.85–5.65) | 0.106 |
| | 35–44 | 159(84.6%) | 29(15.4%) | 3.35(1.44–7.82) | 2.75(1.02–7.4) | 0.045 |
| | ≥45 | 18(62.1%) | 11(37.9%) | 1 | 1 | |
| Agro ecology | Highland | 264(94.3) | 16(5.7%) | 5.258(2.73–10.13) | 7.71(3.72–15.99) | <0.001 |
| | Midland | 164(70.1%) | 70(29.9%) | 0.747(0.45–1.24) | 1.017(0.57–1.8) | 0.955 |
| | Lowland | 91(91.5%) | 29(24.5%) | 1 | 1 | |
| Marital status | Single | 60(74.1%) | 21(25.9%) | 2.857(0.75–10.86) | 1.01(0.81–5.68) | 0.987 |
| | Married | 440(83.7%) | 86(16.3%) | 5.116(1.45–18.05) | 1.53(0.31–7.8) | 0.608 |
| | Divorced | 14(82.4%) | 3(17.6%) | 4.667(0.804–23) | 1.95(0.28–16.05) | 0.53 |
| | Widowed | 5(50%) | 5(50%) | 1 | 1 | |
| Bank account | Yes | 151(88.3%) | 20(117%) | 1.29 (0.73–2.29) | 0.648(0.363–1.16) | 0.142 |
| | No | 368(79.5%) | 95(20.5%) | 1 | 1 | |
| Radio | Yes | 316(87.5%) | 45(12.5%) | 2.42(1.60–3.66) | 1.87(1.17–2.99) | 0.009 |
| | No | 203(74.4%) | 70(24.6%) | 1 | 1 | |
| Mobile phone | Yes | 391(84.1%) | 72(15.6%) | 1.82(1.19–2.29) | 1.92(1.74–3.16) | 0.01 |
| | No | 128(74.9%) | 43(25.1%) | 1 | 1 | |
| Home gardening | Yes | 405(83%) | 83(17%) | 1.37(0.87–2.17) | 0.612(0.332–1.16) | 0.134 |
| | No | 114(78.1%) | 32(21.9%) | 1 | 1 | |
| farm land | Yes | 439(83.5%) | 87(16.5%) | 1.766(1.08–2.88) | 0.66(0.29–1.5) | 0.322 |
| | No | 80(74.1%) | 28(25.9%) | 1 | 1 | |
| Small livestock | Yes | 275(84.1%) | 52(15.9%) | 1.37(0.91–2.05) | 1.38(0.85–2.24) | 0.19 |
| | No | 244(79.5%) | 63(20.5%) | 1 | 1 | |
| large livestock | Yes | 471(84.1%) | 89(15.9%) | 1.77(1.08–2.88) | 2.67(1.4–5.08) | 0.009 |
| | No | 48(64.9%) | 26(35.1%) | 1 | 1 | |
| Average monthly income (ETB)* | ≤500 | 98(97.7%) | 25(20.3%) | 0.37(0.17–0.80) | 0.517(0.21–1.25) | 0.144 |
| | 501–1500 | 314(79.7%) | 80(20.3%) | 0.37(0.18–0.22) | 0.55(0.26–1.19) | 0.133 |
| | ≥1501 | 107(91.5%) | 10(8.5%) | 1 | 1 | |
| Food exchange | Yes | 489(82.6%) | 103(17.4%) | 1.90(0.94–3.8) | 1.62(0.68–3.86) | 0.28 |
| | No | 30(71.4%) | 12(28.6%) | 1 | 1 | |
| Animal transport | Yes | 90(87.4%) | 13(12.6%) | 1.65(0.89–3.06) | 0.79(0.39–1.6) | 0.5 |
| | No | 429(80.8%) | 102(19.2%) | 1 | 1 | |
| Food own farm land | Yes | 483(84%) | 92(16%) | 3.4(1.9–5.9) | 1.8(0.85–3.82) | 0.25 |
| | No | 36(61%) | 23(39%) | 1 | 1 | |
| Food from market | Yes | 469(83.3%) | 94(16.7%) | 0.477(0.27–0.83) | 0.1.178(0.69–3.13) | 0.317 |
| | No | 50(70.4%) | 21(29.6%) | 1 | 1 | |
| Women's decision-making power to purchase food for household | Yes | 426(85.2%) | 74(14.8%) | 3.026(1.92–4.75) | 3.93 (2.3–6.71) | <0.001 |
| | No | 78(65.5%) | 41(34.5%) | 1 | 1 | |

* Average exchange rate in 2018:1$ = 27.6677 ETB.

## Conclusion

The proportion of adequate dietary diversity is high in the sampled population of the Jeldu district, West Shoa zone, Oromia, Ethiopia. Women's making decisions, Agroecological zones, having large livestock, having a radio, and having mobile phones were the predictors of

women's dietary diversity. So health extension workers who are working at the grass root level of the community, district health office, and other concerned bodies should give special attention to the predictors of dietary diversity among women of reproductive age because it improves their dietary diversity.

## Supporting information

**S1 Dataset. Dataset–SPSS.**
(SAV)

## Acknowledgments

The authors would like to thank all respondents for their willingness to participate in the study. They are also grateful to Ambo University College of Medicine Health Sciences, department of Public Health, for material support.

## Author Contributions

**Conceptualization:** Gudisa Merga.

**Data curation:** Gudisa Merga, Nagasa Dida.

**Formal analysis:** Gudisa Merga, Nagasa Dida.

**Funding acquisition:** Gudisa Merga, Gina Kennedy.

**Investigation:** Gudisa Merga, Nagasa Dida.

**Methodology:** Gudisa Merga, Samson Mideksa, Nagasa Dida.

**Project administration:** Gudisa Merga, Gina Kennedy.

**Resources:** Gina Kennedy.

**Software:** Gudisa Merga, Samson Mideksa, Nagasa Dida.

**Supervision:** Gudisa Merga, Samson Mideksa, Nagasa Dida, Gina Kennedy.

**Writing – original draft:** Samson Mideksa.

**Writing – review & editing:** Gudisa Merga, Nagasa Dida, Gina Kennedy.

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
