## [Decision Letter · Decision Letter 0]

10 Jul 2022

PONE-D-22-08826Dietary Diversity and Associated Factors among Reproductive Age Women in Jeldu District, West Shoa Zone, Oromia EthiopiaPLOS ONE

Dear Dr. Dida,

Thank you for submitting your manuscript to PLOS ONE. After careful consideration, we feel that it has merit but does not fully meet PLOS ONE’s publication criteria as it currently stands. Therefore, we invite you to submit a revised version of the manuscript that addresses the points raised during the review process.

Dear authors please appropriately address each reviewers queries. As the questions and comments are many you are required to give sufficient attention to respond Good luck 

We look forward to receiving your revised manuscript.

Kind regards,

Amene Abebe Kerbo, Ph.D.

Academic Editor

PLOS ONE

Journal Requirements:

"Food Systems for Healthier Diets (FSHD) flagship in the CGIAR Research Program on Agriculture for Nutrition and Health (A4NH) for funding this research."

Reviewers' comments:

Reviewer's Responses to Questions

**Comments to the Author**

1. Is the manuscript technically sound, and do the data support the conclusions?

Reviewer #1: Yes

Reviewer #2: Yes

Reviewer #3: Yes

Reviewer #4: Yes

Reviewer #5: Yes

Reviewer #6: Partly

2. Has the statistical analysis been performed appropriately and rigorously? 

Reviewer #1: Yes

Reviewer #2: Yes

Reviewer #3: Yes

Reviewer #4: Yes

Reviewer #5: Yes

Reviewer #6: No

3. Have the authors made all data underlying the findings in their manuscript fully available?

Reviewer #1: Yes

Reviewer #2: No

Reviewer #3: Yes

Reviewer #4: Yes

Reviewer #5: Yes

Reviewer #6: No

4. Is the manuscript presented in an intelligible fashion and written in standard English?

Reviewer #1: No

Reviewer #2: No

Reviewer #3: No

Reviewer #4: No

Reviewer #5: Yes

Reviewer #6: No

5. Review Comments to the Author

Reviewer #1: Manuscript Number: PONE-D-22-08826

Full Title of the Title of paper: Dietary Diversity and Associated Factors among Reproductive Age Women in Jeldu, District, West Shoa Zone, Oromia Ethiopia.

Comments forwarded by the reviewer

The title and abstract cover the main aspect of the work but some issues should be included

Good if the title had time frame

1. Abstract section

1.2 Methods: good to indicate P value used to declare statistical significance.

1.3 Results part of the abstract: equal to or greater or equal to five food shows repetitions of words (need correction of words). It should be relevant to the reader and contain the proportion of reproductive age women who had an overall good/ adequate dietary variety practice, rather than reporting the proportion of reproductive age women who obtained >=5 categories. It's also crucial to present the minimum dietary diversity score as a mean (SD). Furthermore, there is no need to interpret the results in the abstract section; instead, it is preferable to describe the predictor factors that have a significant association with their AOR and 95 percent CI.

1.3 Conclusions part of the abstract: inconsistency (mobile phones and possession of large cattle) It wasn't stated whether or not relevant predictors are present in the abstract's result section, however it did appear in the abstract's conclusion section.

2. Background: the introduction provides background and information relevant to the study, but some issues should be corrected/explained.

Line 63: While defining Dietary diversity it says, the number of foods consumed but it should be the number of food groups consumed not the number of food consumed.

Line 84-89: To relate to the outcome variables, the Author indicated the following target groups: pregnant and breastfeeding women, young children, adolescent females, pregnant, and lactating women. However, they should, in my opinion, keep (specifically) to the target audience specified in the headline (so need consistency). Because the aforementioned groups require a unique study strategy and specific consideration when dealing with dietary diversity.

The authors did not provide a clear magnitude on the problem (at least in the study area), instead focusing on malnutrition. While dietary diversity is a proxy indication of nutritional status, it is good to focus on dietary diversity (specific to title) rather than malnutrition. For example, what is the magnitude/prevalence of reproductive-age women consuming five or more food groups from the list? Or, in Ethiopia, what percentage of reproductive-age women ate a diet that was less varied or monotonous? It should also detail the variables (related factors) that may have contributed to the target group's poor or good dietary diversity practices.

3.Method: Experiments, statistics, and other analyses are performed to a high technical standard and are described in sufficient detail?

In my opinion Authors, tries to do their best, but to be clear and replicable, the methods need certain explanation

Line 104: P (prevalence) Authors used were 50% (i.e. 50% of women of reproductive age obtained adequate dietary diversity. It should be supported with reference.

Line 107: According the Authors report, ten kebeles were selected by lottery method. But it is not clear that from how many Kebeles these 10 kebeles were selected? And what was the assumption to select the given number of Kbeles?

Data collection tools and procedures should be clearly described.

Line 108-109: As a sampling frame, the community health information system (CHIS) was used. Why not use simple random sampling instead of systematic random sampling, which is the most recommended method, if there is a sample frame?

Line 115The authors noted that the questionnaire was written in English, translated into Afan Oromo, and then re-translated into English to ensure consistency. But who is responsible for translating the tools? It should be clarified that this should be done by a language expert.

How will the women's privacy be respected during the interview? What was the location of the interview and how was it conducted? Other family members, for example, did not have free access to the interview location because safeguarding the women's privacy is vital.

Do the Authors have supervisor(s)? If so should be mentioned, the number, their discipline and their activities.

The reader will be able to identify/ capture the relevant independent (predictors) variables that the Authors employed in this study and how those factors were measured to assess the main outcome variable if the major components (or contents) of the questionnaire (or instrument) are provided. This section, in my opinion, should be well stated, as it determines the quality of the research.

Line 121-127: the authors used ten different food groups recommended by the Food and Agricultural Organization to measure DDS. Food groups were assessed using a 24-hour recall method, created as a summary measure of dietary intake, and dichotomized into DDS of (less than five and five and above) to determine whether women had adequate or inadequate dietary diversity. That was the list of good points, but in addition to that, good if

1. The authors should list the ten food groups that were utilized to assess DDS.

2. Make clear whether the authors validated the questionnaire/tool after conducting the assessment of locally available foodstuffs with the mentioned ten food groups according to FAO

3. Specify whether the Authors are evaluating/asking about food just or beverages and/or snacks as well.

4. Make it obvious whether the interviewers are probing the respondents for any food varieties that they may have forgotten.

5. It should be clarified what the authors did if the foods mentioned by the respondents were not in the predetermined list..

Line 128 -132: In Data processing and analysis, I missed the descriptive statistics like mean and Standard Deviation for (continuous variable) and frequency and percentage for (categorical data).

4. Results: The study presents the results of primary scientific research? The research finding produces contribution to the specific vulnerable group. But need clarification in certain areas.

Line 145-146 the authors mentioned that mean of DDS was used, good to include the SD (standard deviation as well). That is (mean ±SD).

Good to give clues for factors which shows significant association with dietary diversity at bivariate analysis and those variables adjusted for. Each variable which shows significant association at multivariate binary logistic regression should be re written again to be self-explanatory.

Example, reproductive age women who were living in highland were 7.71 times more likely to consume a diversified diet compared to their counter parts/or women living in lowland areas [(AOR =7.71; 95% CI: 3.72, 15.99)] and etc.

5. Conclusions: are presented in an appropriate fashion and are supported by the data.

Yes

Results reported have not been published elsewhere? In my understanding not published anywhere

6. Discussions:

160-161: The authors state that the study's nature explains the high proportion of adequate dietary diversity in the study area (i.e. study period). But how certain is this to be the case? It would be better, in my opinion, to rewrite because the likely reason could be a study period. However, that is not the only explanation; there could be others. It's also worth noting if there are any research findings from other study locations that show a higher proportion of adequate DDs than the Jaldu district in Ethiopia.

Line 194: This study's limitation is social desirability bias and recall bias during data collection, which is a reasonable concern raised by the authors, but what the authors did to minimize social desirability bias, particularly recall bias. To reduce recall bias, it is now advised to use a multiple pass 24 hour recall method with good probing techniques.

Line 199: To enhance dietary diversity among reproductive women of reproductive age, the authors suggested that the relevant body pay special attention to predictors of dietary diversity. However, because this is a general recommendation, the relevant entity should be designated and a specific task should be assigned to take action.

7. The article is presented in an intelligible fashion and is written in Standard English?

The manuscript needs language and grammar editions.

8. The research meets all applicable standards for the ethics of experimentation and research integrity?

Yes

Reviewer #2: I would like to thank the authors for the nice work entitled “Dietary Diversity and Associated Factors among Reproductive Age Women in Jeldu District, West Shoa Zone, Oromia Ethiopia” aimed to assess dietary diversity and associated factors among reproductive age women. The clarity in the methodology and the result section were interesting. I have listed my concerns as follows:

1. Methods L 115- Here you have described to have used a tool after reviewing different document, can you please include the references including for the FAO document.

2. Methods L 115- You have mentioned the data collection tool is translated, can you please be specific with the translation process, if it’s translated by a certified language expert.

3. Methods L 123- You have stated that you used a 24-h recall method to assess the food consumed but you have not mention for how many days you implemented the technique. Please include the number of days.

4. Results L140- You have mentioned the average income of the participants to be about between 500-1500 birr. In my opinion it would sound better if you put this amount in USD, for ease of understanding to the readers. In what category of income is this amount considered in the Ethiopian context? Is it a Low, Middle or High income?

5. Results L 149-50- You have included the results for Women’s making decision, Agro ecological zone, having large livestock, having radio and mobile phone. Yet, in the methods section you have not described the justification for choosing these factors are considered appropriate for measuring/determining/ dietary diversity. Please provide your justification for including these factors instead of other possible options.

6. Result L150- You have used terms like ‘large livestock’ is large referring to number or size of the animals. Please define terms before using them.

7. Discussion L161- Your justification for adequate DD is the study period. Primarily, study period is a vague phrase is it the season or the length of your study duration? Secondly how about the effect of other factors such as access to the market, availability of food varieties in the area, tradition of the society under study, the feeding practices and trends etc.? Please try to consider the possible social, economic or environmental factors in your discussion.

8. Discussion L161- The results of this study indicated that the highlanders consume adequate DD is 7.7 more likely than lowlanders. But when referring to the inconsistency with a study in South Tigray you have stated it’s because of the mountainous topography, I am afraid to say but the topography of Southern Tigray is not just mountains but a plain and fertile land and the agricultural products in there ranges from crops to fruits. I believe you can cite other possible reason for that the difference such as variety of crops in the areas or the income level or the access to roads and market or education or any other factor? Or you can cite a reference for your justification.

9. Discussion L176- You have mentioned that women possessing radio/mobile are more likely to attain adequate DD and you have related it to high income. This will be a more interesting discussion if you could include the income level (500-1500 birr) you have stated in the result section to support your justification.

10. Discussion L190- You claimed to have focused on the most vulnerable group. Can you please elaborate on this by rewriting it as ‘…the most vulnerable group such as …’ since it is open to interpretations. The vulnerable group can be attributed to age (children, aged), sex, physical and economic condition etc.

11. Conclusion: Your conclusion is weak and it requires writing it very well.

Minor

1. Abstract L41- Please remove the repeated word.. reproductive “…undertaken among 634 reproductive women of reproductive age.” scientific names should be italicized

2. Abstract L48- re write the sentence “Women those who were living…” as Women who were living

3. Background L95- In the sentence “…mothers' lack of resistance to” please either complete the sentence or remove the ‘to’ at the end.

4. Methods L 111- Typo error, change If to if

5. Methods L 1120- I am not sure if this is a standard abbreviation ‘32hh’ or you should write it in full at first as ‘32 household’

6. I have observed that throughout the manuscript you have capitalized the word women ‘Women’. I didn’t find it appropriate please check it.

7. The grammatical problem is quite high, for instance read the lines between L149-L155 in the last paragraph of the result section. In my opinion your manuscript need professional editing to solve the language issues.

Reviewer #3: Comments to and questions authors

Question1. What is the difference between your study populations, I.e. reproductive age Women and adolescents and pregnant women? As there are, multiple studies conducted in your locality on the mentioned topics what is the contribution of your study to scientific record?

https://doi.org/10.1186/s40985-020-00137-2

https://doi.orgi /10.1186/s40795-017-0148-0

https://doi.org/10.1155/2019/3916864

and others more:

Line 4 - Reproductive women of reproductive age? Do you mean those who gave birth?

Line 104- As you use 50% of proportion, site the literature, If simple consideration why?

Line 105- why design effect? Make it clear

Line 131-make your terms consistent (multivariable logistic regression, multiple logistic regression and multivariate)

Line 132-indicate your level of confidence

General comment;

1. as you developed your questionnaire, the Validity of your question would have checked, but you did not said anything, so I think your tool miss validity

2. There are numerous punctuation problems (spacing, full stop, comma… You need to read detail and correct), long and jargon sentences, check and rewrite it

3. Use standard English language

Reviewer #4: • There are a lot of typographical errors in this manuscript. The language and logic flow of ides should be maintained

• If all the comments are incorporated , I belief the article is of great importance in contributing to scientific evidence

Reviewer #5: Dietary Diversity and Associated Factors among Reproductive Age Women

Manuscript Number: PONE-D-22-08826

1. Please avoid bold letters in abstract section of your Manuscript.

2. I suggest rather than saying study subject use study participants.

3. General comment on Introduction part. This part needs rewriting. In this part you have to write the magnitude, and severity of problem related to poor dietary diversity among Reproductive Age Women starting globally to Ethiopian case. E.g, what problems among Reproductive Age Women due to poor dietary diversity seen in the world it might be in terms of heath problem, growth impairment, loss of productivity or death? What has been done to alleviate the situation and what is not done? This part has to show clear part of the identified gap in available evidences. You can then put your objectives in the study context by justifying its originality and importance. .

4. In line 107, ten kebeles were selected. Please define kebeles for research communities, If it is a population ward, a village or else?

5. Why did you stratified the district into agro-ecological zone? Was the population heterogeneous?

6. You have selected households using systematic random sampling by having sampling frame from community health information system (CHIS). How can you justify that under result section your response rate was 100%?

7. In line 122-127, you have stated that Food consumed by women of reproductive age was assessed through 24-h recall method and then food items were categorized into ten food groups and adequacy of dietary diversity was calculated by summation. I am in doubt if you have you considered the amount or size of food consumed as it was recall method.

8. Where is the operational definition for Adequate Dietary diversity? I suggest if you added it to your manuscript.

9. Reference please for this citation: food and Agriculture Organization (FAO); FHI 360. Minimum Dietary Diversity for Women in line 113.

10. Mention the reference for the statement in line 130 “ variables those having a p ≤0.20 were candidate for multiple logistic regression.

11. In line 216, what was the reason for taking Verbal consent from study subjects aged eighteen years and up?

12. Under result section, only fig.1 uploaded. But you mentioned as table 1, 2, 3. You must upload or remove from referencing in the manuscript.

13. Under your discussion part you said that “This study also confirmed that almost all (99.8%) women consumed grains, root and tubers and this result was supported with the study conducted in Ghana, Zambia; In Kenya and Ethiopia.” Can you analyse the factors associated for this opportunity from your findings as well as others previous studies?

14. Women who owning large livestock were 2.67 times more likely consumed adequate dietary diversity as stated in line 182. By saying large livestock, do you mean size or number? Clarify or operationalize!

15. Regarding average monthly income in Ethiopian birr in line 140, how did you calculated and is it standard for international researchers?

16. In line 152, you mentioned that Women who making decision had adequate dietary diversity. It was better to describe it in terms of household purchase, seeking health service and visiting relatives, who are educated, have a regular earning, owns household or land singly or jointly.

17. In conclusion section, you must reveal the findings of sampled population of the Jeldu district rather than the whole district. Rewrite it!

The end

Thank you very much!

Reviewer #6: Reviewer 1

Comments to the Author

The topic for this article is an interesting and important one. It presents information on dietary diversity and associated factors among children aged 6 - 23 months in Ethiopia. This is useful information; however, the reviewers have raised some important issues that deserve attention.

I suggest the authors identify the gap in the current literature a little bit more clearly, even for Ethiopia since a number of studies are already available. Furthermore, the generalizability of the present study to other populations is a question. It should become clear what is the added value of this study beyond Ethiopia, especially considering that the type of study is not very innovative. Likewise, your results could be compared to other parts of the world.

The manuscript does not bring any novelty to the field, and it uses incorrect terms, making it difficult to understand the study. Hence justify for the above sentence?

Comments

1. In abstract…p value: for binary and multivariate should be sated

2. In abstract-

-Under result, is there highland and highland place in Jaldu district which is stated by CSA Ethiopia? If yes, could u show for us the reference?

Your result in abstract is different from the factor associate witten under line 149-155 ’’- The logistic regression output showed that Women’s making decision, Agro ecological zone having large livestock, having radio and mobile phone significantly associated with dietary diversity women who were living in highland were 7.71 (AOR =7.71; 95% CI: 3.72,15.99) , women who have radio were1.87(AOR= 1.87;95% ;1.17,2.99), Women who making decision were 3.93(AOR =3.93 ; 95% CI: 3.93,2.3,6.71) , women who had large livestock were 2.67( 154 AOR;2.67: 95% CI;1.4,5.08) and women who have mobile phone1.92 (AOR=1.92:95% 155 CI;1.74,3.16,) had adequate dietary diversity (Table 3).’’ Could you justify or correct this thing?

3. Is it reference -1 is book or report or article? Could revise its citation?

4. I think the bracket of PLOS citation according to plos guideline is like= [ ]

Methods

5. Change to methods=method and materials….line 100

6. Write subtitle under method and materials like=Study design, setting, and population….etc until quality control

7. How 10 kebeles were selected? And from how many kebeles?

8. Why you want to use 1.5 design effects as far as u want to generalize your finding for one district or Jaldu district, not west shoa zone.

9. Correct capital to small letter “If” ….line 111

Result

10. Break a sentence in to two and make a sense…. from 134-137

11. There is no variables in table 1, which you wrote here under result ‘’More than (53.6%) of the respondents were protestant in religion, about 83% were 138 married, 55.2% participants, were had no formal education. 57.1% respondents were farmers 139 and 50.6% participants living with less than five family member(table.1).”

12. Make a subtitles under result(socio demographic,factors associated….)

13. On table three under age category(35—44) it shows as it was associated(AOR;2.75(1.02,7.4),P= 0.045),Hence why u did not discussed?

Discussion

14. One line 161 ‘’ The high proportion of adequate dietary diversity in this study is due to the nature of the study i.e study period’’. Justify deeply with citation why the prevalence was high?

15. On line 172-176, ’’The difference is due to agro ecological zone, production availability and socio cultural determinants, most of the highland areas are in Tigray Ethiopia are mountainous and prone to soil erosion and degradation; these could be risk factors for a reduction in agricultural productivity when variety of production are not existing an individual cannot consumed varieties of food groups’’, the question here is, justfy how your study period got high food diversity at highland than lowland as far as most of the highland areas are in Tigray Ethiopia are mountainous and prone to soil erosion and degradation; these could be risk factors for a reduction in agricultural productivity?

6. PLOS authors have the option to publish the peer review history of their article (what does this mean?). If published, this will include your full peer review and any attached files.

Reviewer #1: No

Reviewer #2: **Yes: **Mohammed Hussen Bule

Reviewer #3: No

Reviewer #4: **Yes: **Elias Teferi Bala

Reviewer #5: No

Reviewer #6: No

---

## [Author Response · Author response to Decision Letter 0]

28 Sep 2022

Author's response to Reviewers Comments

PONE-D-22-08826

Title of Manuscript: Dietary Diversity and Associated Factors among Reproductive Age Women in Jeldu District, West Shoa Zone, Oromia Ethiopia

Authors: 

1. Gudisa Merga: gudisamerga2017@gmail.com

2. Samson Mideksa: samkmwmtj@gmail.com

3. Nagasa Dida: nadibefe@yahoo.com

Generally, in manuscript we have used track change to incorporate or address the reviewers’ comments.

A Point by point response

Response to Reviewer #1:

RESPONSE TO SPECIFIC COMMENTS: 

 The title and abstract cover the main aspect of the work but some issues should be included

 Good if the title had time frame

Response from the authors: As suggested by the reviewer, time frame is added to the title.

1. Abstract section 

1.2 Methods: good to indicate P value used to declare statistical significance.

Response from the authors: We have indicated as suggested by the reviewer.

1.3 Results part of the abstract: equal to or greater or equal to five food shows repetitions of words (need correction of words). 

Response from the authors: Corrected

It should be relevant to the reader and contain the proportion of reproductive age women who had an overall good/ adequate dietary variety practice, rather than reporting the proportion of reproductive age women who obtained >=5 categories. 

Response from the authors: ??

It's also crucial to present the minimum dietary diversity score as a mean (SD). Furthermore, there is no need to interpret the results in the abstract section; instead, it is preferable to describe the predictor factors that have a significant association with their AOR and 95 percent CI.

Response from the authors: corrected.

1.3 Conclusions part of the abstract: inconsistency (mobile phones and possession of large cattle) It wasn't stated whether or not relevant predictors are present in the abstract's result section, however it did appear in the abstract's conclusion section.

Response from the authors: In the result section of the abstract having mobile phone is incorporated as one predictor variable 

2. Background: the introduction provides background and information relevant to the study, but some issues should be corrected/explained.

Response from the authors:

Line 63: While defining Dietary diversity it says, the number of foods consumed but it should be the number of food groups consumed not the number of food consumed. 

Response from the authors: Corrected to food groups

Line 84-89: To relate to the outcome variables, the Author indicated the following target groups: pregnant and breastfeeding women, young children, adolescent females, pregnant, and lactating women. However, they should, in my opinion, keep (specifically) to the target audience specified in the headline (so need consistency). Because the aforementioned groups require a unique study strategy and specific consideration when dealing with dietary diversity.

The authors did not provide a clear magnitude on the problem (at least in the study area), instead focusing on malnutrition. While dietary diversity is a proxy indication of nutritional status, it is good to focus on dietary diversity (specific to title) rather than malnutrition. For example, what is the magnitude/prevalence of reproductive-age women consuming five or more food groups from the list? Or, in Ethiopia, what percentage of reproductive-age women ate a diet that was less varied or monotonous? It should also detail the variables (related factors) that may have contributed to the target group's poor or good dietary diversity practices.

Response from the authors: food dietary diversity study among pregnant women was not done before and other area Ethiopia, that is why we use even a proportion of 50% to determine the sample size!

3.Method: Experiments, statistics, and other analyses are performed to a high technical standard and are described in sufficient detail?

In my opinion Authors, tries to do their best, but to be clear and replicable, the methods need certain explanation.

Response from the authors: 

Line 104: P (prevalence) Authors used were 50% (i.e. 50% of women of reproductive age obtained adequate dietary diversity. It should be supported with reference.

Response from the authors: If there is no study done on the same population, statistically we are forced to use an assumption of the proportion of 50% because it gives the maximum sample. 

Line 107: According the Authors report, ten kebeles were selected by lottery method. But it is not clear that from how many Kebeles these 10 kebeles were selected? And what was the assumption to select the given number of Kbeles?

Response from the authors: Total number of the kebles in the districts are included into the abstract and 10 of them were selected with the assumption that at least 30% of the kebeles must be addressed.

Data collection tools and procedures should be clearly described.

Response from the authors: We made revision as indicated in the manuscript using track change. 

Line 108-109: As a sampling frame, the community health information system (CHIS) was used. Why not use simple random sampling instead of systematic random sampling, which is the most recommended method, if there is a sample frame?

Response from the authors: Actually right. If there is sampling frame simple random sampling has to be applied. But, for simplicity and to overcame difficult topography of the district we applied systematic random sampling. 

Line 115The authors noted that the questionnaire was written in English, translated into Afan Oromo, and then re-translated into English to ensure consistency. But who is responsible for translating the tools? It should be clarified that this should be done by a language expert.

Response from the authors: We have used two language experts who translated to English by the first person and re-translated the back to English by the second person.

How will the women's privacy be respected during the interview? What was the location of the interview and how was it conducted? Other family members, for example, did not have free access to the interview location because safeguarding the women's privacy is vital.

Do the Authors have supervisor(s)? If so should be mentioned, the number, their discipline and their activities.

The reader will be able to identify/ capture the relevant independent (predictors) variables that the Authors employed in this study and how those factors were measured to assess the main outcome variable if the major components (or contents) of the questionnaire (or instrument) are provided. This section, in my opinion, should be well stated, as it determines the quality of the research.

Line 121-127: the authors used ten different food groups recommended by the Food and Agricultural Organization to measure DDS. Food groups were assessed using a 24-hour recall method, created as a summary measure of dietary intake, and dichotomized into DDS of (less than five and five and above) to determine whether women had adequate or inadequate dietary diversity. That was the list of good points, but in addition to that, good if

1. The authors should list the ten food groups that were utilized to assess DDS.

Response from the authors: We have put the references that clearly show and list those ten food groups. Moreover, in the result section under Fig. 1 we have listed the ten food groups with their result. 

2. Make clear whether the authors validated the questionnaire/tool after conducting the assessment of locally available foodstuffs with the mentioned ten food groups according to FAO

Response from the authors: From the very beginning we have used the standard tool -FAO 2016. So, we assumed validating the questionnaire is not important.

3. Specify whether the Authors are evaluating/asking about food just or beverages and/or snacks as well.

Response from the authors: Of course, As far as the study concerned with the 24 hr recall, the food they used in 24 hours including snacks were evaluated. 

4. Make it obvious whether the interviewers are probing the respondents for any food varieties that they may have forgotten.

Response from the authors: Perfectly, the respondents were propped for the any food varieties. See the questionnaire we have used.

5. It should be clarified what the authors did if the foods mentioned by the respondents were not in the predetermined list…

Response from the authors: The ten food groups can handle any food, and we haven’t encountered any. But, if we encountered we had been excluded.

Line 128 -132: In Data processing and analysis, I missed the descriptive statistics like mean and Standard Deviation for (continuous variable) and frequency and percentage for (categorical data).

Response from the authors: incorporated as suggested.

4. Results: The study presents the results of primary scientific research? The research finding produces contribution to the specific vulnerable group. But need clarification in certain areas.

Response from the authors: We have indicated that the data were directly collected from the respondents, so that the study is an Original research.

Line 145-146 the authors mentioned that mean of DDS was used, good to include the SD (standard deviation as well). That is (mean ±SD).

Response from the authors: SD is included as suggested by the reviewer.

Good to give clues for factors which shows significant association with dietary diversity at bivariate analysis and those variables adjusted for. 

Response from the authors: list of those variables identified by the bivariable binary logistic regression for the final model were included as indicated using track change.

Each variable which shows significant association at multivariate binary logistic regression should be re written again to be self-explanatory.

Example, reproductive age women who were living in highland were 7.71 times more likely to consume a diversified diet compared to their counter parts/or women living in lowland areas [(AOR =7.71; 95% CI: 3.72, 15.99)] and etc.

Response from the authors: 

5. Conclusions: are presented in an appropriate fashion and are supported by the data.

Yes

Results reported have not been published elsewhere? In my understanding not published anywhere

6. Discussions:

160-161: The authors state that the study's nature explains the high proportion of adequate dietary diversity in the study area (i.e. study period). But how certain is this to be the case? It would be better, in my opinion, to rewrite because the likely reason could be a study period. However, that is not the only explanation; there could be others. It's also worth noting if there are any research findings from other study locations that show a higher proportion of adequate DDs than the Jaldu district in Ethiopia.

Response from the authors: 

Line 194: This study's limitation is social desirability bias and recall bias during data collection, which is a reasonable concern raised by the authors, but what the authors did to minimize social desirability bias, particularly recall bias. To reduce recall bias, it is now advised to use a multiple pass 24 hour recall method with good probing techniques.

Response from the authors: To void the social desirable bias, the data collector informed the respondents about the objective of the stud; by calling the possible time of food consumption time (breakfast, snack, lunch and dinner) in a day the recall bias was minimized as much as possible.

Line 199: To enhance dietary diversity among reproductive women of reproductive age, the authors suggested that the relevant body pay special attention to predictors of dietary diversity. However, because this is a general recommendation, the relevant entity should be designated and a specific task should be assigned to take action.

Response from the authors: corrected by making it specific.

7. The article is presented in an intelligible fashion and is written in Standard English?

The manuscript needs language and grammar editions.

Response from the authors: Revision has made throughout the whole document.

8. The research meets all applicable standards for the ethics of experimentation and research integrity?

Yes

Response to Reviewer #2 comments

Reviewer #2: I would like to thank the authors for the nice work entitled “Dietary Diversity and Associated Factors among Reproductive Age Women in Jeldu District, West Shoa Zone, Oromia Ethiopia” aimed to assess dietary diversity and associated factors among reproductive age women. The clarity in the methodology and the result section were interesting. I have listed my concerns as follows:

1. Methods L 115- Here you have described to have used a tool after reviewing different document, can you please include the references including for the FAO document.

Response from the authors: There references from which tool is adopted is included.

2. Methods L 115- You have mentioned the data collection tool is translated, can you please be specific with the translation process, if it’s translated by a certified language expert.

Response from the authors: It is alright that the translation was made by the language experts, and we have incorporated it.

3. Methods L 123- You have stated that you used a 24-h recall method to assess the food consumed but you have not mention for how many days you implemented the technique. Please include the number of days.

Response from the authors: included as suggested by the reviewer.

4. Results L140- You have mentioned the average income of the participants to be about between 500-1500 birr. In my opinion it would sound better if you put this amount in USD, for ease of understanding to the readers. In what category of income is this amount considered in the Ethiopian context? Is it a Low, Middle or High income?

Response from the authors: the currency exchange of Birr in dollar during the study period is incorporated. 

5. Results L 149-50- You have included the results for Women’s making decision, Agro ecological zone, having large livestock, having radio and mobile phone. Yet, in the methods section you have not described the justification for choosing these factors are considered appropriate for measuring/determining/ dietary diversity. Please provide your justification for including these factors instead of other possible options.

Response from the authors: 

6. Result L150- You have used terms like ‘large livestock’ is large referring to number or size of the animals. Please define terms before using them.

Response from the authors: add under the operational definition.

7. Discussion L161- Your justification for adequate DD is the study period. Primarily, study period is a vague phrase is it the season or the length of your study duration? Secondly how about the effect of other factors such as access to the market, availability of food varieties in the area, tradition of the society under study, the feeding practices and trends etc.? Please try to consider the possible social, economic or environmental factors in your discussion.

Response from the authors: In this study the term study period refers to the time of data collection and the other points are considered.

8. Discussion L161- The results of this study indicated that the highlanders consume adequate DD is 7.7 more likely than lowlanders. But when referring to the inconsistency with a study in South Tigray you have stated it’s because of the mountainous topography, I am afraid to say but the topography of Southern Tigray is not just mountains but a plain and fertile land and the agricultural products in there ranges from crops to fruits. I believe you can cite other possible reason for that the difference such as variety of crops in the areas or the income level or the access to roads and market or education or any other factor? Or you can cite a reference for your justification.

Response from the authors: 

9. Discussion L176- You have mentioned that women possessing radio/mobile are more likely to attain adequate DD and you have related it to high income. This will be a more interesting discussion if you could include the income level (500-1500 birr) you have stated in the result section to support your justification.

Response from the authors: Unfortunately, the average monthly income of the household didn’t show association with the study variable, that is why we haven’t consider for the discussion.

10. Discussion L190- You claimed to have focused on the most vulnerable group. Can you please elaborate on this by rewriting it as ‘…the most vulnerable group such as …’ since it is open to interpretations. The vulnerable group can be attributed to age (children, aged), sex, physical and economic condition etc.

Response from the authors: The vulnerable group to be addressed in this section is to mean the pregnant women, not about the other vulnerable group.

11. Conclusion: Your conclusion is weak and it requires writing it very well.

Response from the authors: Revision has made

Minor

1. Abstract L41- Please remove the repeated word.. reproductive “…undertaken among 634 reproductive women of reproductive age.” scientific names should be italicized

Response from the authors: Removed as indicated.

2. Abstract L48- re write the sentence “Women those who were living…” as Women who were living

Response from the authors: Revised

3. Background L95- In the sentence “…mothers' lack of resistance to” please either complete the sentence or remove the ‘to’ at the end.

Response from the authors: ‘to’ is removed.

4. Methods L 111- Typo error, change If to if

Response from the authors: changed!

5. Methods L 1120- I am not sure if this is a standard abbreviation ‘32hh’ or you should write it in full at first as ‘32 household’

Response from the authors: Right you are it is not standard abbreviation, we have changed hh to household throughout the document

6. I have observed that throughout the manuscript you have capitalized the word women ‘Women’. I didn’t find it appropriate please check it.

Response from the authors: Corrected, except at the beginning of the sentence i.e. if the sentence begin with ‘women’ we have used the upper case.

7. The grammatical problem is quite high, for instance read the lines between L149-L155 in the last paragraph of the result section. In my opinion your manuscript need professional editing to solve the language issues.

Response from the authors: Revision has made across the whole document.

Response to Reviewer #2 comments

Reviewer #3: Comments to and questions authors

Question1. What is the difference between your study populations, I.e. reproductive age Women and adolescents and pregnant women? As there are, multiple studies conducted in your locality on the mentioned topics what is the contribution of your study to scientific record?

https://doi.org/10.1186/s40985-020-00137-2

https://doi.orgi /10.1186/s40795-017-0148-0

https://doi.org/10.1155/2019/3916864

and others more:

Response from the authors: As we all know; reproductive age women are those women who are in the age group of 15 to 49 years; whereas the pregnant women are those women in reproductive age women and pregnant as well. Hence, even if they are in the same age range the population in our case is vast a bit. But, adolescents are those aged from 10 -19 years which need special attention because of physiological development change. Moreover, local information is also valuable for the scientific contribution. 

Line 4 - Reproductive women of reproductive age? Do you mean those who gave birth?

Response from the authors: Corrected, the first reproductive is deleted.

Line 104- As you use 50% of proportion, site the literature, If simple consideration why?

Response from the authors: If there is no study done on the same population, statistically we are forced to use an assumption of the proportion of 50% because it gives the maximum sample. 

Line 105- why design effect? Make it clear

Response from the authors: In sampling procedure; there is stratification and more than sampling techniques is used to select the kebeles (sub-district) and the households; hence, the design effect was used.

Line 131-make your terms consistent (multivariable logistic regression, multiple logistic regression and multivariate) 

Response from the authors: corrected.

Line 132-indicate your level of confidence

Response from the authors: Indicated 

General comment;

1. as you developed your questionnaire, the Validity of your question would have checked, but you did not said anything, so I think your tool miss validity

Response from the authors: Actually, the questionnaire was adapted not developed and to check whether applicable to our context it was pretested. So, we have corrected the term developed to adapted.

2. There are numerous punctuation problems (spacing, full stop, comma… You need to read detail and correct), long and jargon sentences, check and rewrite it

Response from the authors: Done, revision has mad across the whole document.

3. Use standard English language

Response to Reviewer #2 comments

Reviewer #4: • There are a lot of typographical errors in this manuscript. The language and logic flow of ides should be maintained.

Response from the authors: Revision has made as indicated using the track change.

• If all the comments are incorporated , I belief the article is of great importance in contributing to scientific evidence

Reviewer #5: Dietary Diversity and Associated Factors among Reproductive Age Women

Manuscript Number: PONE-D-22-08826

1. Please avoid bold letters in abstract section of your Manuscript.

Response from the authors: Avoided.

2. I suggest rather than saying study subject use study participants.

Response from the authors: Corrected.

3. General comment on Introduction part. This part needs rewriting. In this part you have to write the magnitude, and severity of problem related to poor dietary diversity among Reproductive Age Women starting globally to Ethiopian case. E.g, what problems among Reproductive Age Women due to poor dietary diversity seen in the world it might be in terms of heath problem, growth impairment, loss of productivity or death? What has been done to alleviate the situation and what is not done? This part has to show clear part of the identified gap in available evidences. You can then put your objectives in the study context by justifying its originality and importance. .

Response from the authors: revised

4. In line 107, ten kebeles were selected. Please define kebeles for research communities, If it is a population ward, a village or else?

Response from the authors: It means subdistrict – the smallest administration unit; so we have replaced the kebele with sub-district.

5. Why did you stratified the district into agro-ecological zone? Was the population heterogeneous?

Response from the authors: Right, the population at these different ecological zones harvest different crops, cereals, vegetables and fruit. That means the population have different access to variety of foods. That is why we have stratified the district into ecological zone.

6. You have selected households using systematic random sampling by having sampling frame from community health information system (CHIS). How can you justify that under result section your response rate was 100%?

Response from the authors: One, in the rural area mothers are available at home most of the time unless there is special case. Second, when the target is not available revisit was made. Lastly, the data collection was done out of market day and special events. 

7. In line 122-127, you have stated that Food consumed by women of reproductive age was assessed through 24-h recall method and then food items were categorized into ten food groups and adequacy of dietary diversity was calculated by summation. I am in doubt if you have you considered the amount or size of food consumed as it was recall method.

Response from the authors: you are right, we haven’t assessed the amount. Because Fanta food groups tool doesn’t assess the amount, rather the variety. 

8. Where is the operational definition for Adequate Dietary diversity? I suggest if you added it to your manuscript.

Response from the authors: Operational definition is included. 

9. Reference please for this citation: food and Agriculture Organization (FAO); FHI 360. Minimum Dietary Diversity for Women in line 113.

Response from the authors: Included.

10. Mention the reference for the statement in line 130 “ variables those having a p ≤0.20 were candidate for multiple logistic regression.

Response from the authors: 

11. In line 216, what was the reason for taking Verbal consent from study subjects aged eighteen years and up?

Response from the authors: All information they provided are verbal and there is no any specimen taken from the study participants. 

12. Under result section, only fig.1 uploaded. But you mentioned as table 1, 2, 3. You must upload or remove from referencing in the manuscript.

Response from the authors: All table are in word document.

13. Under your discussion part you said that “This study also confirmed that almost all (99.8%) women consumed grains, root and tubers and this result was supported with the study conducted in Ghana, Zambia; In Kenya and Ethiopia.” Can you analyse the factors associated for this opportunity from your findings as well as others previous studies?

Response from the authors: ????

14. Women who owning large livestock were 2.67 times more likely consumed adequate dietary diversity as stated in line 182. By saying large livestock, do you mean size or number? Clarify or operationalize!

Response from the authors: what large and small mean in this study is included in the definition.

15. Regarding average monthly income in Ethiopian birr in line 140, how did you calculated and is it standard for international researchers?

Response from the authors: the currency exchange of Birr in dollar during the study period is incorporated. 

16. In line 152, you mentioned that Women who making decision had adequate dietary diversity. It was better to describe it in terms of household purchase, seeking health service and visiting relatives, who are educated, have a regular earning, owns household or land singly or jointly.

Response from the authors: Corrected. 

17. In conclusion section, you must reveal the findings of sampled population of the Jeldu district rather than the whole district. Rewrite it!

Response from the authors: Corrected; revision has also made.

Reviewer #6: Reviewer 1

Comments to the Author

The topic for this article is an interesting and important one. It presents information on dietary diversity and associated factors among children aged 6 - 23 months in Ethiopia. This is useful information; however, the reviewers have raised some important issues that deserve attention.

I suggest the authors identify the gap in the current literature a little bit more clearly, even for Ethiopia since a number of studies are already available. Furthermore, the generalizability of the present study to other populations is a question. It should become clear what is the added value of this study beyond Ethiopia, especially considering that the type of study is not very innovative. Likewise, your results could be compared to other parts of the world.

The manuscript does not bring any novelty to the field, and it uses incorrect terms, making it difficult to understand the study. Hence justify for the above sentence?

Response from the authors: Corrected. 

Comments

1. In abstract…p value: for binary and multivariate should be sate

Response from the authors: stated. 

2. In abstract-

-Under result, is there highland and highland place in Jaldu district which is stated by CSA Ethiopia? If yes, could u show for us the reference?

Response from the authors: 

Your result in abstract is different from the factor associate witten under line 149-155 ’’- The logistic regression output showed that Women’s making decision, Agro ecological zone having large livestock, having radio and mobile phone significantly associated with dietary diversity women who were living in highland were 7.71 (AOR =7.71; 95% CI: 3.72,15.99) , women who have radio were1.87(AOR= 1.87;95% ;1.17,2.99), Women who making decision were 3.93(AOR =3.93 ; 95% CI: 3.93,2.3,6.71) , women who had large livestock were 2.67( 154 AOR;2.67: 95% CI;1.4,5.08) and women who have mobile phone1.92 (AOR=1.92:95% 155 CI;1.74,3.16,) had adequate dietary diversity (Table 3).’’ Could you justify or correct this thing?

Response from the authors: Corrected. 

3. Is it reference -1 is book or report or article? Could revise its citation?

Response from the authors: As indicated under the reference, it is a five site study result summary and it is recommended to site as ‘Mary Arimond, et al. Dietary Diversity as a Measure of the Micronutrient Adequacy of Women’s Diets In Resource-Poor Areas: Summary of Results from Five Sites. Washington, DC: FANTA-2 Bridge, FHI 360, 2011.’

4. I think the bracket of PLOS citation according to plos guideline is like= [ ]

Response from the authors: Corrected. 

Methods

5. Change to methods=method and materials….line 100

Response from the authors: changed. 

6. Write subtitle under method and materials like=Study design, setting, and population….etc until quality control

Response from the authors: subtitle are give as suggested by the reviewer.. 

7. How 10 kebeles were selected? And from how many kebeles?

Response from the authors: indicated. 

8. Why you want to use 1.5 design effects as far as u want to generalize your finding for one district or Jaldu district, not west shoa zone.

Response from the authors: In sampling procedure; there is stratification and more than sampling techniques is used to select the kebeles (sub-district) and the households; hence, the design effect was used.

9. Correct capital to small letter “If” ….line 111

Response from the authors: Corrected.

Result

10. Break a sentence in to two and make a sense…. from 134-137

Response from the authors: Separated.

11. There is no variables in table 1, which you wrote here under result ‘’More than (53.6%) of the respondents were protestant in religion, about 83% were 138 married, 55.2% participants, were had no formal education. 57.1% respondents were farmers 139 and 50.6% participants living with less than five family member (table.1).”

Response from the authors: Table was missed form the table list, the socio-demographic factor table is added.

12. Make a subtitles under result (socio demographic factors associated….)

Response from the authors: Subtitle are given.

13. On table three under age category (35—44) it shows as it was associated (AOR;2.75(1.02,7.4), P= 0.045), Hence why u did not discussed?

Response from the authors: Discussed. 

Discussion

14. One line 161 ‘’ The high proportion of adequate dietary diversity in this study is due to the nature of the study i.e study period’’. Justify deeply with citation why the prevalence was high?

Response from the authors: Justified as indicated on the manuscript using the track change. 

15. On line 172-176, ’’The difference is due to agro ecological zone, production availability and socio cultural determinants, most of the highland areas are in Tigray Ethiopia are mountainous and prone to soil erosion and degradation; these could be risk factors for a reduction in agricultural productivity when variety of production are not existing an individual cannot consumed varieties of food groups’’, the question here is, justfy how your study period got high food diversity at highland than lowland as far as most of the highland areas are in Tigray Ethiopia are mountainous and prone to soil erosion and degradation; these could be risk factors for a reduction in agricultural productivity? 

Response from the authors: Revision has made.

---

## [Editor Report · Decision Letter 1]

3 Oct 2022

PONE-D-22-08826R1Dietary Diversity and Associated Factors among Reproductive Age Women in Jeldu District, West Shoa Zone, Oromia Ethiopia, 2018PLOS ONE

Dear Dr. Dida,

Thank you for submitting your manuscript to PLOS ONE. After careful consideration, we feel that it has merit but does not fully meet PLOS ONE’s publication criteria as it currently stands. Therefore, we invite you to submit a revised version of the manuscript that addresses the points raised during the review process.

We look forward to receiving your revised manuscript.

Kind regards,

Amene Abebe Kerbo, Ph.D.

Academic Editor

PLOS ONE
---

## [Author Response · Author response to Decision Letter 1]

2 Dec 2022

We have no words for the reviewers, the have made us. The comments they have provided are very valuable and we have learnt a lot from it. Many thanks to all of them.

---

## [Editor Report · Decision Letter 2]

5 Dec 2022

Dietary Diversity and Associated Factors among Women of Reproductive Age in Jeldu District, West Shoa Zone, Oromia Ethiopia

PONE-D-22-08826R2

Dear Mr Negasa

We’re pleased to inform you that your manuscript has been judged scientifically suitable for publication and will be formally accepted for publication once it meets all outstanding technical requirements.

Kind regards,

Amene Abebe Kerbo, Ph.D.

Academic Editor

PLOS ONE
---

## [Editor Report · Acceptance letter]

8 Dec 2022

PONE-D-22-08826R2 

Dietary Diversity and Associated Factors among Women of Reproductive Age in Jeldu District, West Shoa Zone, Oromia Ethiopia 

Dear Dr. Dida:

I'm pleased to inform you that your manuscript has been deemed suitable for publication in PLOS ONE. Congratulations! Your manuscript is now with our production department. 

Kind regards, 

on behalf of

Dr. Amene Abebe Kerbo 

Academic Editor

PLOS ONE